# The Use of Mirror Therapy in Peripheral Seventh Nerve Palsy: A Systematic Review

**DOI:** 10.3390/brainsci14060530

**Published:** 2024-05-23

**Authors:** Marco Castaldo, Giovanni Sellitto, Ilaria Ruotolo, Anna Berardi, Giovanni Galeoto

**Affiliations:** 1Department of Anatomical, Histological, Forensic and Orthopaedic Sciences, Sapienza University of Rome, 00185 Rome, Italy; 2MS Center, S. Andrea Hospital, Sapienza University, 00185 Rome, Italy; giovanni.sellitto@uniroma1.it; 3Department of Human Neurosciences, Sapienza University, 00185 Rome, Italy; ilaria.ruotolo@uniroma1.it (I.R.); anna.berardi@uniroma1.it (A.B.); 4IRCCS Neuromed, 86077 Pozzilli, Italy

**Keywords:** peripheral facial nerve palsy, Bell’s palsy, mirror therapy

## Abstract

Background: Conservative therapy is currently the elective treatment for peripheric facial palsy according to scientific literature. The success of conservative therapy is due to physiotherapy and the application of its methods. The aim of this systematic review was to assess mirror therapy, a physiotherapeutic method. Objectives: The aim of the following systematic review is to evaluate the effectiveness of using mirror therapy in patients with peripheral paralysis of the seventh cranial nerve. Methods: This systematic review was conducted according to Preferred Reporting Items for Systematic Reviews and Meta-analyses (PRISMA) guidelines. The screening of literature was carried out on Cochrane, PEDro, PubMed/Medline, Scopus and Web of Science databases up until August 2022. All studies were randomized controlled trials (RCTs) and 5 articles met the inclusion criteria and were included in this study. The risk of bias was evaluated with PEDro and Jadad scales. Discussion: In the present study, we reviewed 5 RCTs that compared mirror therapy with other physiotherapy treatments or placebo to reduce pain, depression and improve range of motion in patients with peripheric facial nerve palsy. Conclusions: Further studies are needed to determine the effectiveness of this type of treatment, but nevertheless the data obtained are very encouraging.

## 1. Introduction

The facial nerve is the seventh cranial nerve. Motor fibers leave the facial nerve as the greater petrosal nerve. Peripheral paralysis of the seventh cranial nerve is the most frequent cranial neuropathy and can originate from various types of damage to the seventh nerve including its motor nucleus. Bell’s palsy is the most prevalent peripheral paralysis of the seventh cranial nerve and has a rapid unilateral onset. The diagnosis is one of exclusion and is most often made on objective examination. The facial nerve follows an intracranial, infratemporal and extratemporal progression and the same occurs to its branches. It performs a motor and parasympathetic function controlling upper and lower facial muscles and the taste for the front two-thirds of the tongue. It also controls the salivary glands and lacrimal ones. For diagnosis, it is necessary to pay special attention to forehead muscle strength: if it is preserved, it should be considered a central cause of weakness [1].

There are two types of facial paralysis, the central one due to lesions involving the descending bundles directed to the facial nerve nuclei and the peripheral ones due to lesions of the facial nucleus, roots, and nerve trunk [2].

The annual global incidence of Bell’s palsy is 15–30 cases per 100,000 people and the lifetime risk is 1 in 60 [3,4]. There is a recurrence rate of 8% to 12%. It can appear at any age, but more cases are noted in middle and late age, with the median occurrence at 40 years of age. Risk factors include diabetes, pregnancy, pre-eclampsia, obesity, and hypertension [1,5].

Without intervention, 71% of patients with idiopathic facial palsy have complete recovery after 1 year, 13% have mild residual weakness, and 16% have fair to poor recovery. The advantages of early intervention include increased patient education in the disease process, proper eye care and initial exercises at home. Given the likelihood of spontaneous recovery in idiopathic causes of facial palsy, the optimal treatment for this condition remains controversial [6].

There are conflicting opinions on the use of physiotherapy for the treatment of Bell’s palsy. Some suggest that it can be useful for patients because it helps maintain facial muscle tone and helps stimulate the nerve (National Institute of Neurological Disorders and Stroke 2003). This is supported by Beurskens and Heymans (2003), whose research identified a significant improvement in facial movement in those who received mime therapy (a combination of mime and physiotherapy) [7].

Mirror therapy (MT) is commonly defined as a rehabilitation therapy in which a mirror is placed between the arms or legs so that the image of a moving non-affected limb gives the illusion of normal movement in the affected limb. By this setup, different brain regions for movement, sensation, and pain are stimulated. Mirror therapy applied within facial nerve palsy rehabilitation consists of facial neuromuscular retraining, principles of motor learning, and motor imagery sessions [8,9].

The aim of this systematic review is to evaluate the effectiveness of using mirror therapy in patients with peripheral paralysis of the seventh cranial nerve in order to indicate clinical lines of intervention based on the scientific literature [10]. The secondary objective is to assess the methodological quality of the studies included in the review.

## 2. Material and Methods

This study was not registered on Prospero. The systematic review was conducted according to Preferred Reporting Items for Systematic Reviews and Meta-analyses (PRISMA) guidelines of 2020 [11].

### 2.1. Eligibility Criteria

To be included in this systematic review, papers had to report the results of randomized controlled trials (RCTs). Studies published in English were chosen and examined the effectiveness of mirror therapy alone or in combination with other treatments for people with peripheral facial nerve palsy.

### 2.2. Information Sources and Search Strategy

The literature research was performed on PubMed (via MEDLINE), Scopus (via EBSCO), PEDro, and Cochrane databases and Web Of Science using the following keywords: ((“peripheral facial nerve palsy”) OR (“Bell’s palsy”) AND (“mirror therapy”)); the final time of revision is 31 December 2023.

### 2.3. Inclusion Criteria

The following inclusion criteria were followed to assess the studies: (1) patients with a diagnosis of peripheral paralysis of the VII cranial nerve; (2) reference only to mirror therapy or comparison with placebo or other techniques; (3) articles published in English; (4) full text available; and (5) 18 years since publication.

Reviews, case reports, letters or editorials, and studies that did not meet this criterion were excluded.

### 2.4. Data Collection and Analysis

The following information about the studies were collected: (1) references (authors and year of publication); (2) participants (number, age, diagnosis); (3) duration of the study; (4) presence of follow-up; (5) intervention; (6) control; (7) scales for outcome evaluation; and (8) results/conclusions.

### 2.5. Risk of Bias (RoB)

The RoB was evaluated using one of the available adequate instruments for the included studies. The Jadad scale, Pedro scale and Risk of bias 2 tool were applied to each of the included studies; a table of risk of bias is included in the published review, with evidence to support each judgment. The authors did not use the Jadad scale and Pedro scale for non-randomized study designs because these tools are used to assess the methodological quality of RCTs; instead, they assessed the evidence’s validity as part of the interpretation of results [12,13].

RoB assessment was implemented through the Cochrane RoB 2 tool for RCTs, following the Cochrane Handbook for Systematic Reviews of Interventions [14]. The tool has five different domains used to generate the “Overall RoB”. The RoB judgment for the second domain (RoB due to deviations from planned interventions) was carried out to quantify both the effect of the assignment to the intervention and the effect of starting and adhering to intervention. The third and fourth domains of the RoB-tool (RoB due to missing outcome data and RoB in measurement of the outcome) were quantified instead on each of the measures of outcome present in the works included in the revision. Each domain was evaluated with one of the following options: “Low RoB”, “Some Concerns”, and “High RoB”. The criteria used for the evaluation of the RoB of the studies follow the Cochrane directives, based upon which they are judged “Low RoB”. The studies that presented for all domains with low RoB are instead judged “Some Concerns”; the studies that have no more than a single domain were also judged “Some Concerns”. The trials were judged to be at high risk of bias in at least one domain of result, or the trial was judged to have some concerns for multiple domains in a way that substantially lowered confidence in the result. RoB for each study was evaluated by two authors and disagreements were resolved by negotiation [15].

### 2.6. Meta-Analysis

The meta-analysis study was conducted on studies that reported comparable follow-ups for comparable outcome measures. Results were combined whenever available in at least two comparable studies. The data analysis was carried out with RevMan Web, a software package developed by the Cochrane collaboration (which is a free software environment for statistics), using the package “meta” (quote RevMAn web). The results were combined using fixed-effect models (DerSimonian–Laird method). The heterogeneity between studies has been assessed with statistics I, which represents the percentage of variance due to heterogeneity rather than chance and is not sensitive to the number of studies involved. The I values ranged from 0% to 100% and were interpreted as low–moderate if less than 50%. The results and the heterogeneity obtained were summarized in tables with the graph “forest plot”.

## 3. Results

An initial literature search (Table 1) through databases identified 79 studies. A total of 75 studies were excluded. In the end, five studies were included (Figure 1). In Table 2, the data of single studies are presented.

### 3.1. Study Characteristics: Types of Design and Types of Participants

Upon the completion of the screening and selection process, five studies had been identified; the identified studies were all randomized controlled trials. The sample size in five of the included studies [6,8,16,17,18] ranged from 20 [16] to 64 [17]. In two studies [16,17], the majority of patients were males; in the remaining study [6], gender had not been specified. Studies use different modalities to indicate the ages of subjects; Table 2 contains this information. Studies include different primary outcome measurement tools:House–Brackmann scale: The standard method for measuring facial nerve function is the use of the House–Brackmann facial nerve rating system, introduced in 1983 (House and Brackmann) and approved by the Committee on Facial Nerve Disorders of the American Academy of Otolaryngology in 1984. This rating scale is designed to accurately describe a patient’s facial function and to monitor its status over time to assess the course of recovery and the effects of treatment. It is also a scale that can be used quickly enough to be useful in clinical practice [19]. The scale analyzes the following: synkinesis, symmetry, rigidity, and global motility of the face. It is divided into 6 categories: normal, mild dysfunction, moderate dysfunction, moderate- severe dysfunction, severe dysfunction, and total paralysis. Its score ranges from 0 to 6, where 6 corresponds to total paralysis [20].Sunnybrook facial grading system: The Sunnybrook facial grading system consists of three domains (face at rest, voluntary movement, synkinesis), three facial regions at rest (eye, cheek, mouth), and five facial regions in voluntary movement with or without synkinesis (eyebrow lift, gentle eye closure, snarl, open-mouthed smile, and lip ripple). Within each domain and facial region, there are three to five levels. The system generates a composite score that describes the overall static and dynamic state of the face. The maximum score is 100 and represents normal facial symmetry [21].FaCE: The FaCE questionnaire is a validated (QoL) instrument that is used to assess facial impairment and disability after facial palsy. It consists of 15 statements, each with five Likert Scale items. A participant circles the most appropriate response to a given statement, whereby 1 corresponds to the lowest function and 5 corresponds to the highest function. These statements are then grouped into six independent domains: social function, facial movement, facial comfort, oral function, visual comfort and tear control. An overall score incorporates all these domains. Using a specific formula, a score is calculated from 0 (worst) to 100 (best) [22].Beck Depression Inventory Scale (BDI) is among the most used self-rating scales for measuring depression worldwide [23].Facial Disability Index: The FDI is a brief, self-reported questionnaire of physical disability and psychosocial factors related to facial neuromuscular function. It is intended to assess disability and the outcome of intervention in terms of meaningful change in the patient’s physical disability and psychosocial status [24].

### 3.2. Synthesis of Evidence

As already mentioned, the objective of this systematic review is to prove the effectiveness of mirror therapy for peripheral paralysis of the seventh cranial nerve. Five RCTs, conducted by various research groups, were analyzed with the common aim of evaluating the effectiveness of this method.

Regarding the study by Paolucci et al. (2020) [16], the aim was to study the effects of mirror therapy by comparing it with conventional rehabilitation, i.e., mimic and myofascial therapy. The analysis of functional assessments demonstrates that both groups underwent a progressive improvement from t0 to t3, and stabilization of the results occurred at follow-up. A significant difference was found in House–Brackmann scale scores between t0 and follow-up, in support of the experimental group. Concerning QoL (assessed through the facE scale), both total scores and social functions improved in both groups from t0 to t3. The experimental group performed better with regard to QoL and emotional depression. The combined use of Mt and Mi with the classical protocol of mime therapy and the myofascial technique is effective in the rehabilitation of the facial nerve palsy (FNP), improving facial physical function. Furthermore, the EG achieved better results regarding QoL and emotional depression. More studies are necessary to determine the predictive factors involved in facial mimicry recovery, with regard to the communicative and empathic features of facial nerve palsy. Finally, the use of mirror therapy has been shown to be effective in the rehabilitation of peripheral paralysis of the seventh cranial nerve.

Mughal et al. (2021) [17] evaluated the effectiveness of facial restraining with and without mirror therapy in people with Bell’s palsy. Visual mirror feedback was used in combination with neuromuscular restraining; this approach was more effective in improving facial symmetry and movement and in reducing functional disability than single use in patients with Bell’s palsy. The study concluded that both treatments seem to be effective in improving Facial Disability Index and House–Brackmann scale scores. However, NMR combined with MVF was found to be more effective in enhancing symmetry and movement in the face, and in decreasing functional disability on the 3rd and 7th week follow-up than NMR used alone in Bell’s palsy patients.

In the study by Martineau et al. (2020) [18], the aim of the study was to provide preliminary evidence about the long-term effects of a new facial training based on motor imagery and mirror therapy. Significant differences were not found between the groups for any measured variable; however, the treatment group experienced better recovery, with reference to all measured variables. This tendency increased for patients with severe or total paralysis. These results showed that a trend toward better symmetry for patients with severe or total and acute BP following early facial retraining. This study provides preliminary clinical evidence that the MEPP could be efficient and can be safely implemented in a clinical trial designed to investigate its efficacy in those with acute severe or total BP.

Martineau et al. (2022) [8] suggest that mirror therapy could promote the recuperation of patients with Bell’s palsy. It could favor the improvement of face symmetry and a decrease in synkinesis in the long term, with a quantifiable impact one year after onset. In brief, the findings of this study imply that mirror therapy within an early facial training protocol (as the MEPP) could support the rehabilitation of patients with acute and severe Bell’s palsy. In particular, early facial training with MEPP could help improvement in facial symmetry and a decrease in synkinesis in the long run, with quantifiable impact at one year after onset. MEPP could also help people to reach a better quality of life (QoL) than counseling alone (when it is provided in the acute phase). According to observers, patients with severe facial palsy did not show intelligibility issues, and the MEPP did not considerably change intelligibility ratings compared to controls. In future studies, a larger sample with more patients rated with severity grades 5 and 6 on the House–Brackman 2.0 scale should be enrolled. Further, a direct comparison between MEPP and other interventions (such as conventional individualized facial rehabilitation) would be interesting to conduct. At last, to overcome subjectivity caused by human-generated facial ratings through tools such as House–Brackmann 2.0 or Sunnybrook, it would be interesting to assess facial symmetry with objective instrumental facial metrics tools, for example, emotrics.

### 3.3. Risk of Bias within Study

The Jadad score and Pedro scale were used for the qualitative analysis of the studies included in this systematic review. According to this assessment, it was noted that three studies [8,16,17] obtained a score > 3, revealing high-level quality. The remaining studies [6,18] had a score that indicated a low qualitative level. All data are shown in Table 3 and Table 4.

The detail of the evaluation of the RoB is shown in Figure 2, Figure 3, Figure 4, Figure 5 and Figure 6. On the whole, the opinions were almost all of “High RoB”, with the exception of the work of Mughal et al., [17] who expressed some concerns in domain 2 only with regard to the effect of assignment of intervention.

In detail, it stands out that almost all the tools used to measure the results present in the studies showed a high RoB in measurement of the outcome.

It is important to underline that all authors reported HBS as an outcome. Although it is reported in all the studies included in the review, it was not possible to carry out the meta-analysis as different follow-ups and different types of analyzes are reported. according to the instrument used, all outcomes obtained a high-risk value.

## 4. Discussion

Bell’s palsy is a debilitating condition, particularly for those patients experiencing severe enduring sequelae despite an appropriate medication intake [25,26]; the severity of symptoms is definitely variable, although generally, the prognosis is very positive [27].

Without intervention, 71% of idiopathic facial palsy patients have a complete recovery after 1 year, 13% have a slight residual weakness, and 16% have fair to poor recovery [28].

The most recent systematic review that investigates this issue was carried out by Mughal et al. (2021) [17] and concluded that both treatments (facial neuromuscular retraining with mirror visual feedback) seem to be effective in improving Facial Disability Index and House–Brackmann scale scores, which were found to be correlated with each other in the assessment of QoL for patients with Bell’s palsy [29]. However, NMR combined with MVF was found to be more effective in promoting symmetry and movement of face and in decreasing functional disability on 3rd and 7th week follow up than NMR performed alone in Bell’s palsy patients [30].

Mughal et al. (2021) [17] claims that the patients who receive appropriate medication but still suffer from severe Bell’s palsy after 14 days seem to benefit from the MEPP. The MEPP significantly improved patients’ QoL during recovery and most probably contributed to decreasing synkinesis at one-year post-onset; the MEPP supports recovery of facial symmetry.

The current literature shows different kinds of treatments for facial nerve palsy rehabilitation. Khan et al. conducted a systematic review in 2022 in which evidence for treatment with facial exercise therapy was presented; the most commonly used techniques were facial exercise therapy, physical therapy combined with biofeedback, facial exercise therapy combined with corticosteroids, botulinum toxin, electrical stimulation, and laser treatment. This review supports facial exercise therapy, but to date, information on the specific benefits of therapy at different timepoints post-onset of facial palsy or for patients living with different levels of severity is difficult to understand with certainty [31].

The results that emerged from this systematic review corroborate previous findings; in fact, mirror therapy appears to be more beneficial if started early. We expect mirror therapy to be more beneficial for patients with recruitment in the paretic phase (3–6 months after onset) than in the later phases, especially for recovery of symmetry in facial movements, in addition, mirror therapy appears to be more palatable to patients, who demonstrate much better levels of treatment adherence and willingness to travel to specialized centers than those receiving other therapies [32].

The evaluation of the RoB through the proposed tools showed different evaluations depending on the tools used. Although three studies were of high methodological quality according to the Jadad and Pedro scale, the risk assessment through RoB 2 showed that there is no high methodological quality to the outcomes proposed by the different authors. These data could be caused by the rigorous risk assessment. In fact, for the total risk domain to be positive, all items must have obtained a low risk of bias. To interpret the risk, it is recommended to refer to each domain and not to the total value.

### Study Limits

One limit of this study is the small number of randomized controlled trials in the current literature. The heterogeneity of the condition, the small sample size of subjects included in the trials, and the lack of double-blinding are related to the nature of the included studies. It should be noted that studies included in this systematic review do not detail the clinical variants of peripheral seventh nerve palsy. Finally, the studies’ partial information shows that the different timing of follow-ups and the different outcome measures used by researchers did not allow us to perform a meta-analysis.

## 5. Conclusions

Based on the results of the conducted literature review, it was concluded that mirror therapy influences improvements in facial functional abilities in patients with facial palsy. Due to the low number of experimental studies presented in the current literature, it was difficult to find an answer to our study question.

Further studies are needed to investigate the effectiveness of this kind of treatment, but at the same time, the data obtained are very encouraging.

## Figures and Tables

**Figure 1 brainsci-14-00530-f001:**
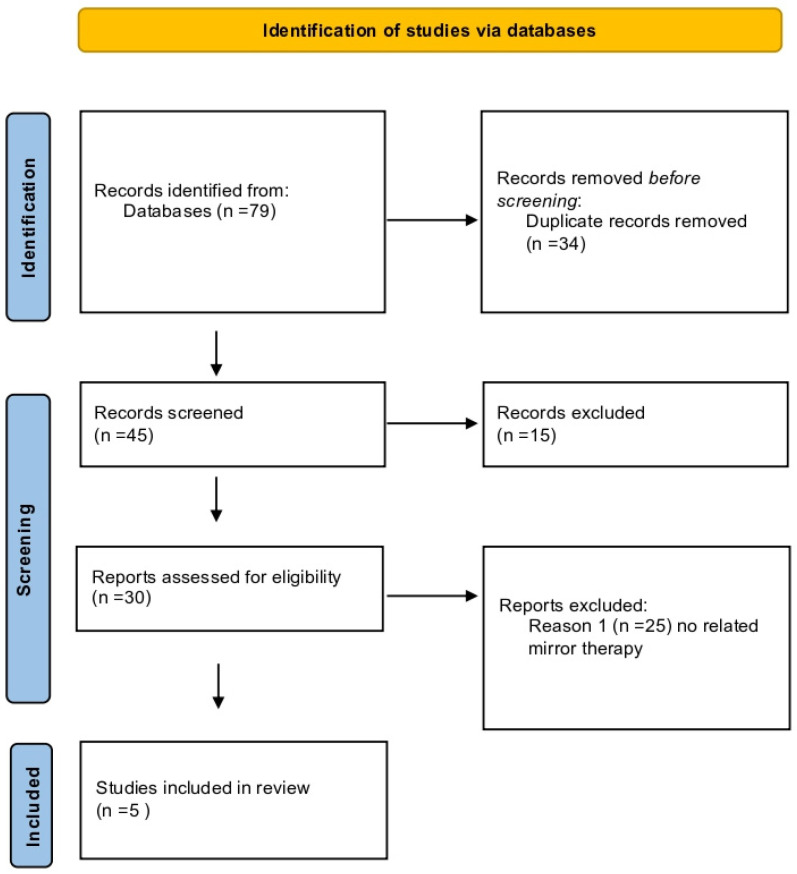
Flowchart of the study.

**Figure 2 brainsci-14-00530-f002:**
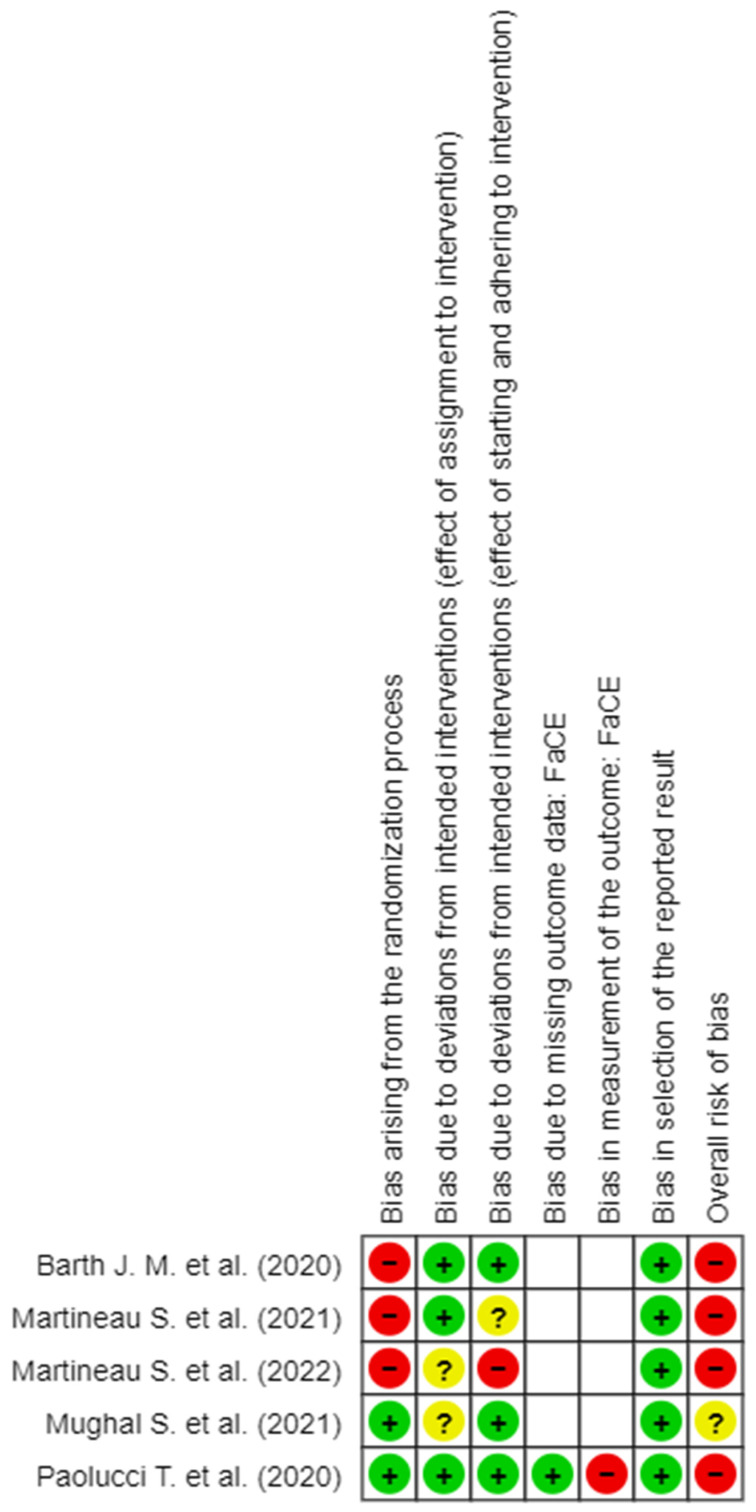
RoB 2 for facE scale [6,8,16,17,18].

**Figure 3 brainsci-14-00530-f003:**
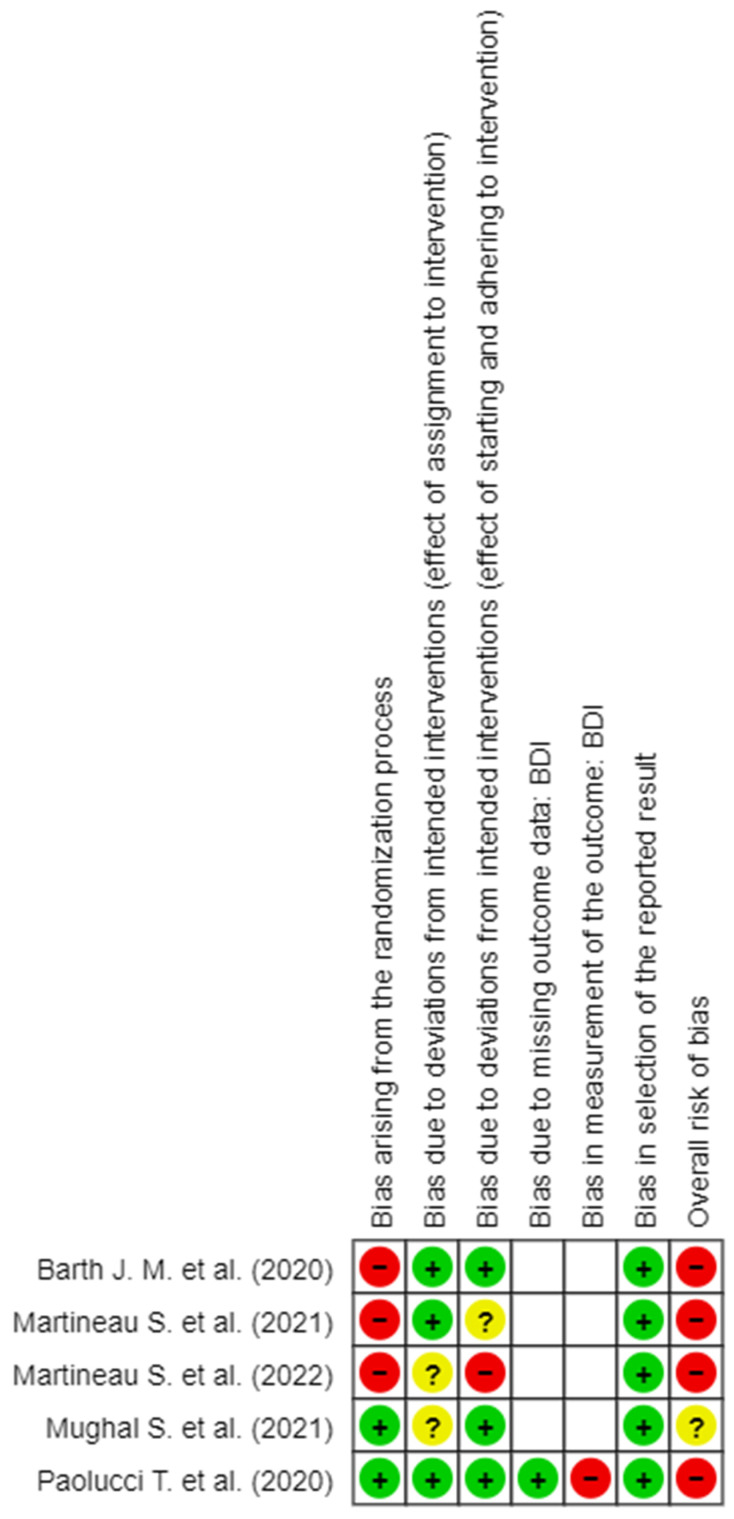
RoB 2 for BDI scale [6,8,16,17,18].

**Figure 4 brainsci-14-00530-f004:**
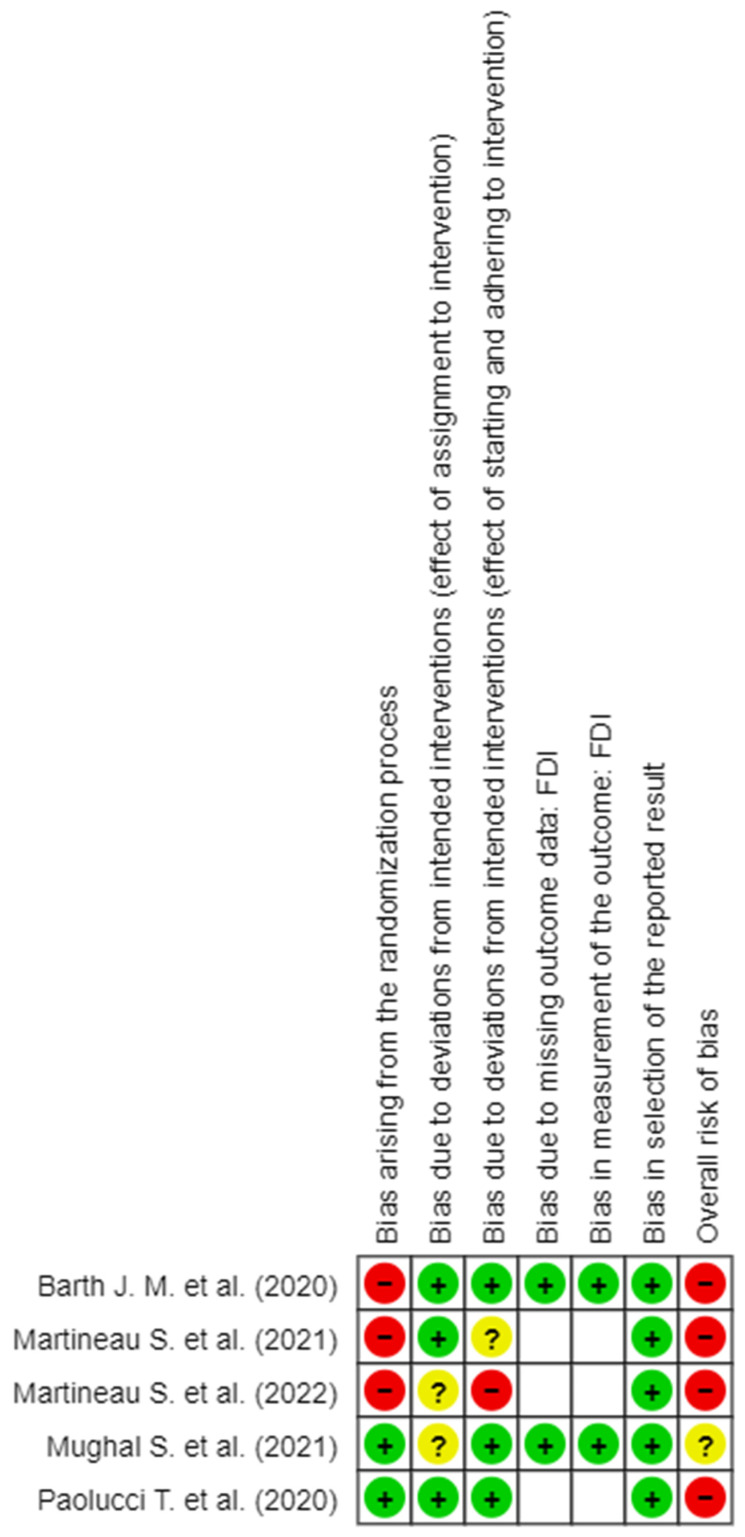
RoB 2 for FDI scale [6,8,16,17,18].

**Figure 5 brainsci-14-00530-f005:**
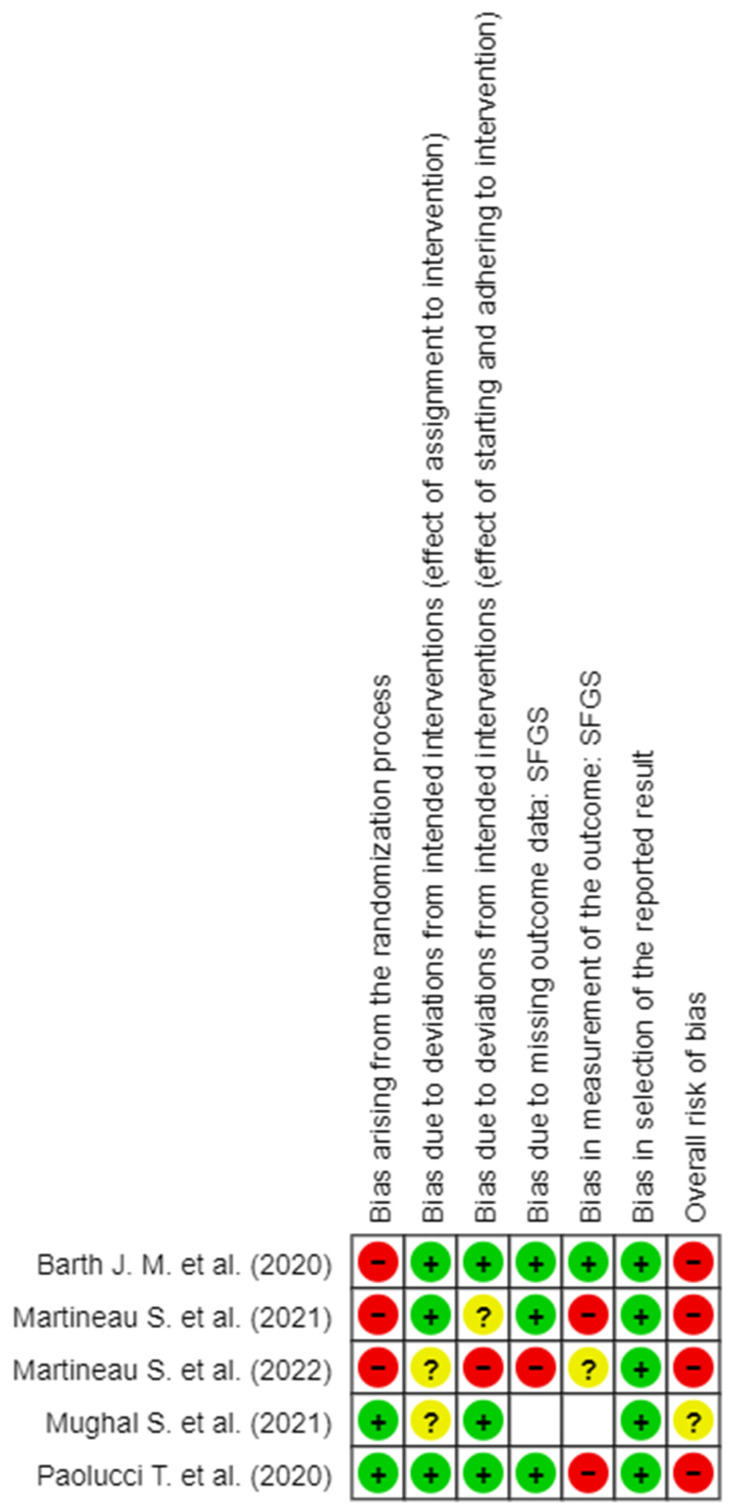
RoB 2 for SFGS scale [6,8,16,17,18].

**Figure 6 brainsci-14-00530-f006:**
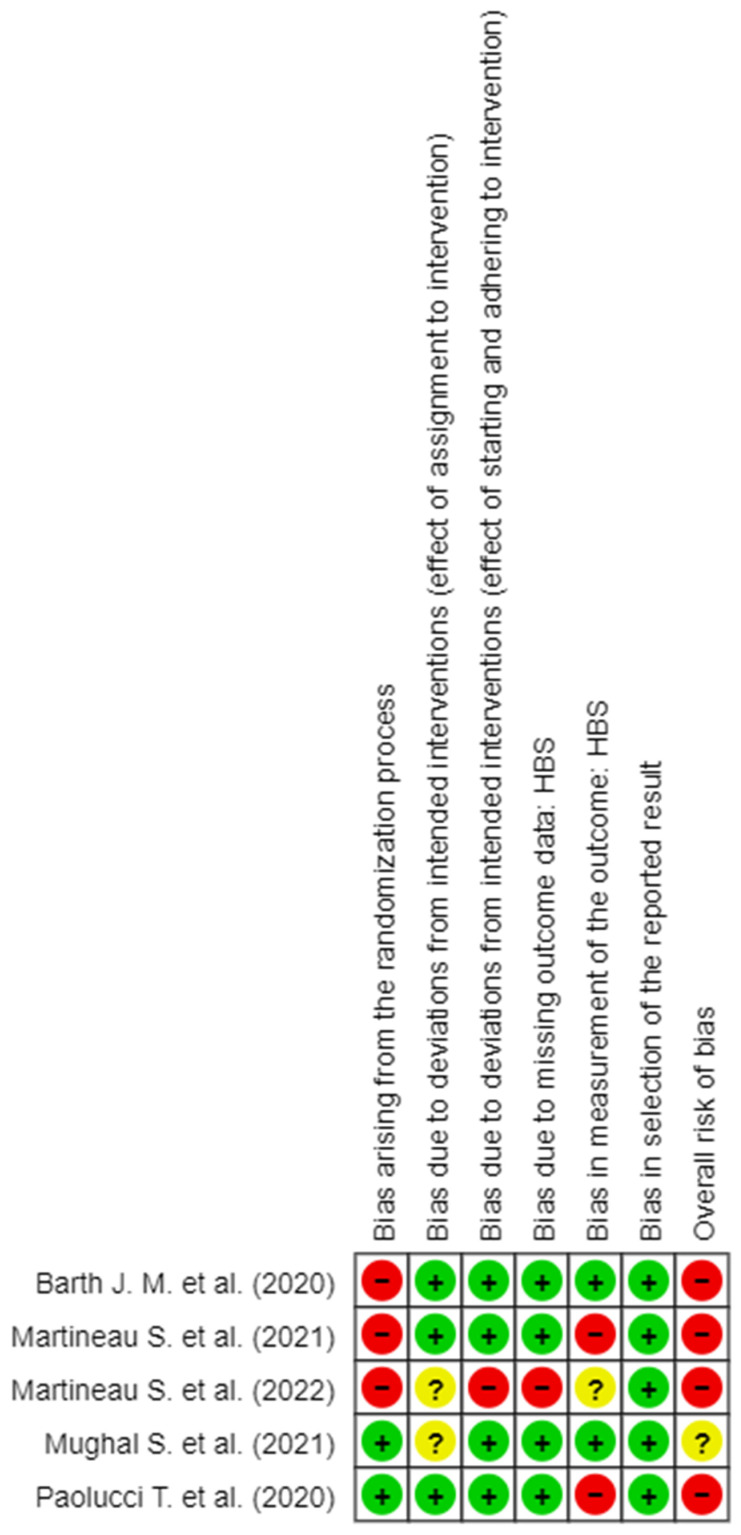
RoB 2 for HBS scale [6,8,16,17,18].

**Table 1 brainsci-14-00530-t001:** Research strategies and number of records.

Database	Key Words	Number of Records
PubMed/MEDLINE	“peripheral facial nerve palsy OR Bell’s palsy AND mirror therapy”	22
Pedro	“peripheral facial nerve palsy OR Bell’s palsy AND mirror therapy”	7
Scopus	“peripheral facial nerve palsy OR Bell’s palsy AND mirror therapy”	21
Cochrane Library	“peripheral facial nerve palsy OR Bell’s palsy AND mirror therapy”	12
Web Of Science	“peripheral facial nerve palsy OR Bell’s palsy AND mirror therapy”	17

**Table 2 brainsci-14-00530-t002:** Characteristics of the study.

Author	Population	Duration of Treatment and Follow-Up	Intervention	Control	Outcome	Results/Conclusions
Paolucci et al. (2020) [16]	20 patients Group I (10 patients) Group II (110 patients) Age group I: 49 Age group II: 48.5 Gender Group I: (5 F; 5M) Group II: (4 F; 6 M)	Outcome assessments were performed before treatment (T0), after one month (T1 = 10 sessions, twice/week), after the second and thirds months (T2 = 10 twice/week + 5 of MT+MI one/week and T3 = 10 twice/week + 5 of MT + MI 1/week), and at the 4-week follow-up (T4 = 2 months follow-up)	In the experimental group, patient performed an additional weekly session of mirror therapy, from the beginning of the rehabilitation, using specific software to create real time symmetric facial images and record data to monitor their rehabilitative treatment and motor imagery exercises	The traditional rehabilitation treatment included mime therapy and then an initial session on information about the treatment and prognosis with the physiatrist; self-massage of the face and neck; breathing and relaxation exercises; exercises to coordinate both sides and reduce synkinesis and for eye and lip closure per the myofascial approach to rehabilitation; letter and word exercises; and expressive exercises	The House–Brackmann scale The Sunnybrook facial grading system The facial clinimetric evaluation scale Beck depression inventory scale	The analysis of the functional evaluations showed that both groups experienced progressive improvement t0 to t3, with stabilization of the results at the follow-up. There was a significant difference in House–Brackmann scale scores between T0 and follow-up in favor of the experimental group. In terms of quality of life (facE scale), total scores and social function items improved in both groups from t0 to t3. The experimental group obtained better results regarding quality of life and emotional depression
Mughal et al. (2021) [17]	64 patients Group I (32 patients) Group II (32 patients) Age group I: 42.06 Age group II: 41.25 Gender Group I: (26 F; 6 M) Group II: (19 F; 13 M)	9 months Follow-up 3rd week Follow-up 7th week	Both groups received neuromuscular retraining exercises (NMR). Group 1 received mirror visual feedback (MVF) additionally	Control group received neuromuscular retraining exercises	Facial disability index and House–Brackmann scale	Mirror visual feedback used in combination with NMR was found more effective in improving the facial symmetry and movement and in decreasing functional disability than NMR used alone in Bell’s palsy patients
Barth et al. (2020) [6]	25 patients Group I (10 patients) Group II (15 patients)	In the mirror book therapy group, patient treatment duration ranged from 2 to 19 sessions. In the standard rehabilitation group, patient treatment duration ranged from 2 to 22 sessions	15 patients received mirror book therapy in conjunction with standard facial rehabilitation	10 patients received facial physical rehabilitation including manual therapy and postural exercises	Facial Grading System (FGS) score, the Facial Disability Index–Physical (FDIP) score, and the Facial Disability Index–Social (FDIS) score	The addition of mirror book therapy to standard facial rehabilitation treatments does significantly improve outcomes in the treatment of idiopathic facial palsy
Martineau et al. (2020) [18]	20 patients Group I (10 patients) Group II (10 patients) Age group I: 50.0 Age group II: 50.1 Gender Group I (6 F; 4 M) Group II (5 F; 5 M)	6 months	The experimental group received the MEPP program (motor imagery + manipulations + facial mirror therapy)	The control group received basic counseling	The House–Brackmann 2.0 scale and the Sunnybrook facial grading system	This study provides preliminary clinical evidence that the MT could be efficient
Martineau et al. (2022) [8]	40 patients Group I (20 patients) Group II (20 patients) Age group I: 48.2 Age group II: 47.9 Gender Group I (8 F; 12 M) Group II (10 F; 10 M)	All patients underwent eight assessments spread over 12 months. The first assessment took place 10–14 days after Bell’s palsy onset and before any facial therapy. Each subsequent assessments were performed at 1, 2, 3, 4, 5, 6, and 12 months after onset	The experimental group received the MEPP program (motor imagery + manipulations + facial mirror therapy).	The control group received basic counseling	The House–Brackmann 2.0 scale and the Sunnybrook facial grading system	Descriptive statistics demonstrated improvements in favor of the MT for each measured variable

**Table 3 brainsci-14-00530-t003:** Risk of bias (Jadad scale).

	Item 1	Item 2	Item 3	Item 4	Item 5	Total Score
Paolucci et al., 2020 [16]	1	1	1	1	1	5
Martineau et al., 2020 [18]	1	1	0	0	1	3
Mughal et al., 2021 [17]	1	1	0	0	0	2
Barth et al., 2020 [6]	1	0	0	0	0	1
Martineau et al., 2022 [8]	1	1	0	0	1	3

**Table 4 brainsci-14-00530-t004:** Risk of bias (PEDRO SCALE).

	Item 1	Item 2	Item 3	Item 4	Item 5	Item 6	Item 7	Item 8	Item 9	Item 10	Item 11	Total Score
Paolucci et al., 2020 [16]	YES	YES	YES	YES	YES	NO	YES	YES	YES	YES	YES	9
Martineau et al., 2020 [18]	YES	YES	YES	YES	NO	NO	YES	YES	YES	YES	YES	8
Mughal et al., 2021 [17]	YES	YES	YES	YES	NO	NO	NO	YES	YES	YES	YES	7
Barth et al., 2020	YES	YES	YES	YES	YES	YES	NO	NO	YES	YES	YES	8
Martineau et al., 2022 [8]	YES	YES	YES	YES	NO	NO	YES	YES	YES	YES	YES	8

## Data Availability

The original contributions presented in the study are included in the article, further inquiries can be directed to the corresponding author.

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
