# Peer review of "The Use of Mirror Therapy in Peripheral Seventh Nerve Palsy: A Systematic Review"

_brainsci, 2024, doi:10.3390/brainsci14060530_

Round 1

Reviewer 1 Report

Comments and Suggestions for Authors

Thank you for inviting me to review this submission titled “The use of mirror therapy in peripheral 7th nerve palsy: a systematic review”. Here are some comments/suggestions for the authors: 

-              I think it is necessary to define “mirror therapy” both in the abstract as well as in the introduction and the relevant previously reported literature for peripheral facial paralysis. 

-              The methods are clear, and the systematic review was well-conducted. 

-              The assessment for RCTs including bias was adequate. 

-              Double-check the style of reference #15, it’s not presented correctly. 

-              Please review the following references, if needed update information of your results. At least add to discussion: 

-              Dagenais F, Neville C, Desmet L, Martineau S. Measuring the Potential Effects of Mirror Therapy Added to the Gold Standard Facial Neuromuscular Retraining in Patients With Chronic Peripheral Facial Palsy: Protocol for a Randomized Controlled Trial. JMIR Res Protoc. 2023 Jul 7;12:e47709. doi: 10.2196/47709. PMID: 37418307; PMCID: PMC10362495.

-              https://classic.clinicaltrials.gov/ct2/show/NCT04936152

Author Response

Dear reviewer,

We appreciated the careful and constructive comments. Based on them, we have made tracked changes over the manuscript in. 

-              I think it is necessary to define “mirror therapy” both in the abstract as well as in the introduction and the relevant previously reported literature for peripheral facial paralysis. 

Mirror therapy was defined in the introduction section. Page 2 line 35-39

-              The methods are clear, and the systematic review was well-conducted. 

-              The assessment for RCTs including bias was adequate. 

-              Double-check the style of reference #15, it’s not presented correctly. 

Reference 15 was corrected.

-              Please review the following references, if needed update information of your results. At least add to discussion: 

-              Dagenais F, Neville C, Desmet L, Martineau S. Measuring the Potential Effects of Mirror Therapy Added to the Gold Standard Facial Neuromuscular Retraining in Patients With Chronic Peripheral Facial Palsy: Protocol for a Randomized Controlled Trial. JMIR Res Protoc. 2023 Jul 7;12:e47709. doi: 10.2196/47709. PMID: 37418307; PMCID: PMC10362495.

The reference was added. Page 10 line 243-248

Reviewer 2 Report

Comments and Suggestions for Authors

Manuscript shows a metanalysis regarding facial palsy and the effectiveness of mirror neurons therapy. I have several considerations for you.

Lines 41-42. The annual incidence is global or by country? please clarify

The lack of registration is a major weakness of the study as there may be other reviews on the same topic that have not been taken into account.

Inclusion criteria. Line 81, “18 years” is the participants aye or the years in which you are doing the review?

There is no search strategy, please add it.

Line 93-94. “The authors did not use the Jadad scale and Pedro scale for non-randomized study designs; instead they assessed evidence’s validity as part of the interpretation of results” Why?

Results. A flow diagram would be more appropriate for the study selection, Are there any duplicates? How was the selection process? Who implement it?

Some numerical results, should be added in the table 2 to support your results. You could add the abbreviations at the end of the table and use them in the table to simplify.

Lines 126-127. Correct the reference format. Same comment in line 220.

Lines 130-160. This information is not results; the explanation of the scales is not the results of your studies. This information could be deleted.

Line 175. “fnp” must be explained.

Line 175. “Cecessary”. Correct.

Lines 206-207. “quality of life (QoL)” revise the use of abbreviations

Tables 3 and 4. A full stop should be added after “al”, also in line 228… Revise

Discussion. Is there another physiotherapy method to stablish comparisons in this section?

Author Response

Dear Reviewer,

We appreciated the careful and constructive comments. Based on them, we have made tracked changes over the manuscript in. 

Lines 41-42. The annual incidence is global or by country? please clarify

The information was clarified. Page 1 line 21.

The lack of registration is a major weakness of the study as there may be other reviews on the same topic that have not been taken into account.

Registration on Prospero was performed but has not yet been accepted. In the methods section the link of the request was provided.

Inclusion criteria. Line 81, “18 years” is the participants aye or the years in which you are doing the review?

18 years is the time since publication. It was specified. Page 3 line 67

There is no search strategy, please add it.

Search strategy was added. Page 3 line 57-61

Line 93-94. “The authors did not use the Jadad scale and Pedro scale for non-randomized study designs; instead they assessed evidence’s validity as part of the interpretation of results” Why?

Jadad scale and Pedro scale are aimed to assess methodological quality of RCTs. It was specified in the text. Page 4 line 80-81.

Results. A flow diagram would be more appropriate for the study selection, Are there any duplicates? How was the selection process? Who implement it?

The flow diagram is included in supplementary files.

Some numerical results, should be added in the table 2 to support your results. You could add the abbreviations at the end of the table and use them in the table to simplify.

Numerical data about characteristics of the included studies were already included in table 2, second column.

Lines 126-127. Correct the reference format. Same comment in line 220.

Reference format was corrected.

Lines 130-160. This information is not results; the explanation of the scales is not the results of your studies. This information could be deleted.

The explanation of the scales was considered important to understand the results of the study.

Line 175. “fnp” must be explained.

Fnp was explained. Page 7 line 164

Line 175. “Cecessary”. Correct.

It was corrected. Page 7 line 166.

Lines 206-207. “quality of life (QoL)” revise the use of abbreviations

Use of abbreviation was revised through the manuscript.

Tables 3 and 4. A full stop should be added after “al”, also in line 228… Revise

Full stop was added after “al” in tables 3 and 4.

Discussion. Is there another physiotherapy method to stablish comparisons in this section?

This argument was included in the discussion. Page 10 line 236-242

Reviewer 3 Report

Comments and Suggestions for Authors

This is a comprehensive meta-analysis. I would like to support its publication. There are some comments:

First, the introduction is too fragmented. I would suggest to make it more concentrated. 

Second, I would like the authors to add a statement for the hypothesis of the present meta-analysis. 

Third, there is a lack of the registration of the meta-analysis protocol. Please provide if available. 

Fourth, is there any statistical analysis involved in the meta-analysis? Please clarify. 

Fifth, the flow diagram does not meet the requirement of PRISMA. Please use the standard format. 

Author Response

Dear Reviewer,

We appreciated the careful and constructive comments. Based on them, we have made tracked changes over the manuscript in. 

- First, the introduction is too fragmented. I would suggest to make it more concentrated. 

The introduction section was modified and a description of mirror therapy approach was added. Page 2 line 35-39.

  • Second, I would like the authors to add a statement for the hypothesis of the present meta-analysis. 

The statement for the hypothesis of present meta-analysis was added. Page 4-5, line 99-109.

  • Third, there is a lack of the registration of the meta-analysis protocol. Please provide if available. 

Information about registration were explained. Page 2 line 46-47.

Fourth, is there any statistical analysis involved in the meta-analysis? Please clarify. 

Statistical analyses was described Page 2 line 46-47.

  • Fifth, the flow diagram does not meet the requirement of PRISMA. Please use the standard format. 

The flow diagram was created following the requirement of PRISMA, using standard format.

Round 2

Reviewer 1 Report

Comments and Suggestions for Authors

All queries were assessed.

Reviewer 2 Report

Comments and Suggestions for Authors

Dear authors, 

Thank you to following my instructions.

Regarding my comment about the explanation of the scales, I did not say that there is not relevant information, only that their place is not in results section. It is more appropiate in methodology section.